# Observation of the ponderomotive effect in non-valence bound states of polyatomic molecular anions

Do Hyung Kang [1], Jinwoo Kim [1], Heung-Ryoul Noh [2] & Sang Kyu Kim [1]✉

The ponderomotive force on molecular systems has rarely been observed hitherto, despite potentially being extremely useful for the manipulation of the molecular properties. Here, the ponderomotive effect in the non-valence bound states has been experimentally demonstrated, for the first time to the best of our knowledge, giving great promise for the manipulation of polyatomic molecules by the dynamic Stark effect. Entire quantum levels of the dipole-bound state (DBS) and quadrupole-bound state (QBS) of the phenoxide (or 4-bromophenoxide) and 4-cyanophenoxide anions, respectively, show clear-cut ponderomotive blue-shifts in the presence of the spatiotemporally overlapped non-resonant picosecond control laser pulse. The quasi-free electron in the QBS is found to be more vulnerable to the external oscillating electromagnetic field compared to that in the DBS, suggesting that the non-valence orbital of the former is more diffusive and thus more polarizable compared to that of the latter.

[1] Department of Chemistry, KAIST, Daejeon 34141, Republic of Korea. [2] Department of Physics, Chonnam National University, Gwangju 61186, Republic of Korea. ✉email: sangkyukim@kaist.ac.kr

The ponderomotive force is the physical phenomenon experienced by the charged particle in the presence of the oscillating electromagnetic fields[1–3], which is extremely useful in the spatiotemporal manipulation of atoms/molecules/ions. The utilization of the ponderomotive force becomes more and more important not only in resolving scientific issues but also in the materialization of the cutting-edge modern quantum technology. As a trigger of the wiggling motion, the ponderomotive force has been vastly employed for the acceleration of electron[4,5] or neutral atoms[6], aligning the atoms in an optical lattice[7,8], the high-harmonic generation[9], or the construction of the trapped-ion quantum simulators[10,11]. The ponderomotive force shares the same origin with the dynamic Stark effect[12–14]. The scientific merit of the dynamic Stark effect, though it is still quite rare, was demonstrated in terms of its active role in chemical reaction control as well as in the spectroscopic characterization of the ionization/detachment processes[15–20]. And yet, it has mostly been confined to atoms or diatomic molecules only. The dynamic Stark effect on the polyatomic molecular system seems to be still in infancy in terms of both the scientific understanding and its chemicophysical applications. From the molecular perspective, this is partially due to the fact that Rydberg states, which are expected to be largely influenced by the ponderomotive force[21–25], are extremely short-lived in the polyatomic molecules due to the strong and complicated interaction with the ionic core.

In this regard, the quasi-free electron residing in the non-valence orbital associated with the polyatomic molecular anions is extremely attractive. As the non-valence excess electron in the non-valence bound state (NBS) of the anion is quite loosely bound through the long-range monopole-multipole[26–29] or correlation interaction[30,31], it is highly anticipated to behave like a free electron in the oscillating electromagnetic field. Since there is only one state for the excess electron in the NBS most cases, the quantum defects caused by the electron-core interaction are expected to be non or insignificant. In this sense, the non-valence orbital in the various NBSs such as dipole-bound state (DBS)[32–34], quadrupole-bound state (QBS)[35–38], or correlation-bound state (CBS)[39–42] seems to be an ideal target for the investigation of the pondermotive effect on the polyatomic molecular anions. It should be emphasized that the intrinsic nature of the monopole-dipole (or quadrupole) or correlation interaction is anticipated to be little influenced by the molecular complexity[43], meaning that the ponderomotive force could be effective even for the quite large polyatomic systems as long as the non-valence orbital survives during the oscillating electromagnetic field. It has been recently found that the DBS prepared below the detachment threshold, unlike the Rydberg states of polyatomic molecules, could survive quite long with the lifetime much longer than tens of nanoseconds or microseconds[44], although it should be noted yet that the NBS lifetime is subject to the detailed electronic structures in terms of the coupling between non-valence and valence orbitals[45–47]. The NBS levels above the detachment threshold mainly decay by the rovibrational autodetachment process. According to our recent real-time dynamics studies on DBS and QBS[48,49], the Fermi-Golden rule provides the nice platform for the description of both absolute and relative autodetachment rates.

Herein, we tackled three different molecular anions (phenoxide (PhO$^-$), 4-bromophenoxide (4-BP$^-$), or 4-cyanophenoxide (4-CP$^-$)) to investigate the ponderomotive force on the excess quasi-free electron in the non-valence orbital of NBS. Quite remarkably, we found that all the non-valence bound states show the clear-cut ponderomotive effect induced by the non-resonant picosecond (ps) laser pulse at 791 nm. The overall behavior of the ponderomotive blue-shift ($\Delta\tilde{\nu}$) with the increase of the laser intensity ($I$)

follows the free-electron model (vide infra). This already gives the important message that the large polyatomic molecular anions could be manipulated through the pondermotive force on the quasi-free electron in the non-valence orbitals. Intriguingly, whereas both DBS and QBS follow the free-electron model in terms of the linearity of $\Delta\tilde{\nu}$ versus $I$, the QBS behaves more like a free-electron than the DBS does. The model potential functions are found to be quite useful for the visualization of the diffuseness and associated polarizability of the excess electron in the non-valence orbital.

## Results and discussion

**Photodetachment spectra with and without the control laser pulse**. The photodetachment spectrum of the cryogenically-cooled (~35 K) PhO$^-$ taken by monitoring the total photoelectron signal as a function of the excitation energy of the ps pump laser pulse ($\Delta t \sim 1.7$ ps, $\Delta E \sim 20$ cm$^{-1}$) is compared with that taken in the presence of the spatiotemporally overlapped non-resonant ps control laser pulse at 791 nm (Fig. 1b). Overall, the sharp bands correspond to the DBS resonances whereas the broad background is due to the direct photodetachment process. The DBS at zero-point energy (ZPE) level is located below the electron affinity (EA) threshold whereas additional sharp vibrational Feshbach resonances are above the EA threshold[50]. The DBS at ZPE was found to survive quite long (» ns)[44], whereas the 11$'^1$ mode decays by the autodetachment process with the lifetime of ~33 ps[48]. At the zero-delay time, the non-resonant (791 nm) ps control laser pulse is spatiotemporally overlapped with the scanning pump laser pulse, and all the DBS bands are found to be blue-shifted, indicating that the entire DBS electronic state is lifted up by the amount of the ponderomotive potential given by the control laser pulse intensity. Each DBS band is also broadened toward the blue-edge of the peak. The stepwise increase of the direct photoelectron signal at the EA threshold exhibits the blue-shift, although the quantitative estimation seems to be nontrivial due to the lack of the sharpness of the step-like structure. Similar to the case of PhO$^-$, the photodetachment spectra of the 4-CP$^-$ QBS also show the same pattern of the blue-shifts for all the QBS bands in the presence of the non-resonant control laser pulse (Fig. 1c). As the electron binding energy of the 4-CP$^-$ QBS is only ~ 20 cm$^{-1}$[36], which is much smaller than that of the PhO$^-$ DBS (~97 cm$^{-1}$)[50,51], the spectral isolation of the ZPE level of the QBS is less straightforward. However, blue-shifts and peak broadenings are quite clearly observed also in the 4-CP$^-$ QBS.

In order to characterize the spatiotemporal circumstances of the ponderomotive force given to the system, we examined the behavior of the most prominent 11$'^1$ DBS band of PhO$^-$ as the delay time between pump and control laser pulses is varied. When the blue edge of the 11$'^1$ DBS band was monitored as a function of the pump-control delay time (Figs. 1 and 2), the Gaussian-shaped transient with the full-width at half-maximum (FWHM) of ~3.3 ps was obtained. This is in excellent agreement with the nominal cross-correlation width (~2.8 ps) of pump and control laser pulses. In Fig. 2, the 11$'^1$ DBS band were obtained at various pump-control delay times by monitoring only the low kinetic-energy electron to avoid the contribution of the control laser pulse. Obviously, both the blue-shift and asymmetric spectral broadening take place, only when the 791 nm control pulse ($I \sim 85$ GW/cm$^2$) is spatiotemporally overlapped with the pump laser pulse. Asymmetric broadening to the blue-edge of the DBS or QBS band with the increase of the control laser intensity is attributed to the Gaussian shapes of the control laser pulse in both temporal and spatial domains. In the weakly overlapped region, the ponderomotive shift becomes small. But as the shift is

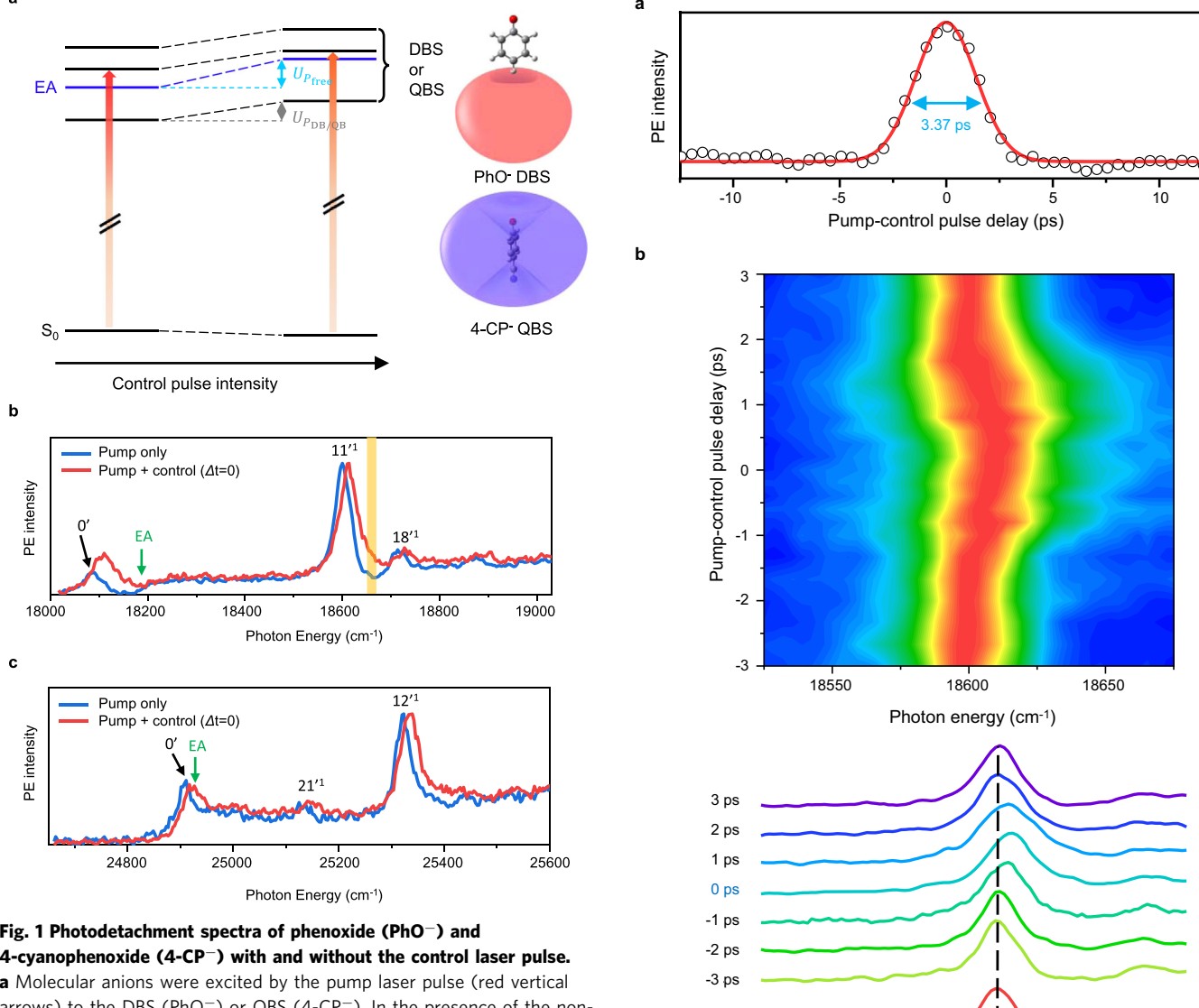

**Fig. 1 Photodetachment spectra of phenoxide (PhO⁻) and 4-cyanophenoxide (4-CP⁻) with and without the control laser pulse.** **a** Molecular anions were excited by the pump laser pulse (red vertical arrows) to the DBS (PhO⁻) or QBS (4-CP⁻). In the presence of the non-resonant control pulse, all the quantum levels of DBS or QBS showed the blue-shift whereas those of ground anions are little affected. Approximate non-valence orbital shapes of the DBS and QBS are depicted. Picosecond photodetachment spectrum of the **b** PhO⁻ and **c** 4-CP⁻ with (red) or without (blue) the non-resonant control pulse (791 nm, ~ 85 GW/cm²) at the zero-time delay. Electron affinity (EA) in the absence of the control pulse is labeled in green. The vibrational mode assignments of the PhO⁻ DBS (or 4-CP⁻ QBS) were denoted on the corresponding peaks (see the text). The yellow bar in **b** denotes the pump laser wavelength where the photoelectron was monitored as a function of the delay time between the pump and control laser pulses (Fig. 2a). Source data are provided as a Source Data file.

always towards the blue-edge, regardless of the spatiotemporal directions, the consequent DBS or QBS band (Supplementary Fig. 1) ends up with the asymmetric broadening which stands out at the blue-edge of the band. One may conceive the combined use of the narrower-bandwidth pump laser pulse and higher-power control laser pulse for the better energy-resolution and larger shift of the ponderomotive effect, respectively, which is subject to the further investigation.

**The ponderomotive shift as a function of the control laser intensity.** In general, the ponderomotive (or dynamic Stark)

**Fig. 2 Temporal (overlap between pump and control laser pulses) behavior of the ponderomotive shift. a** The photoelectron transient taken by fixing the pump laser at the blue-edge of the PhO⁻ DBS 11′¹ peak (denoted by the yellow shaded stick in Fig. 1b) while scanning the time-delay between the pump and control laser pulses. The transient was fitted with a Gaussian function. **b** Photodetachment spectra of the PhO⁻ DBS taken at various delay times between pump and control pulses. The most prominent 11′¹ band (centered at ~ 18,595 cm⁻¹, black dashed line) shows the blue-shift and asymmetric broadening as the temporal overlap becomes maximized. The intensity of the non-resonant ps laser pulse (791 nm) was ~85 GW/cm². Source data are provided as a Source Data file.

potential energy is given as follows.

$$V \approx -\frac{1}{4}\alpha E^2(t) \qquad (1)$$

here, $\alpha$ is the polarizability of the charged particle whereas $E(t)$ is the external oscillating electric field. For the free electron, $\alpha$ is

expressed as[13];

$$\alpha_e(\omega) = -\frac{e^2}{m_e \omega^2} \qquad (2)$$

where $\omega$ is the angular frequency of the control laser pulse. In the classical treatment, therefore, the lift-up energy ($U_p$) from the ponderomotive effect is then given as follows for the free-electron.

$$U_p = \frac{e^2 E^2}{4 m_e \omega^2} \qquad (3)$$

Obviously, this so-called ponderomotive potential ($U_p$) of the free electron is linearly proportional to the laser intensity ($I \propto E^2$). As the driving force for $U_p$ in DBS or QBS is identical, the blue-shift of the $11'^1$ DBS band of PhO$^-$ or the $12'^1$ QBS band of 4-CP$^-$ gives the linear relationship with the laser intensity as expected (Fig. 3 and Supplementary Fig. 2). However, the slopes of $\Delta\tilde{\nu}$ versus $I$ are found to be quite different to each other. Namely, compared to the slope of $\eta = (|\alpha_e|/2\epsilon_0 c)$ for the free-electron model from Eq. (3), the linear fit to the experiment gives the slope of $(0.26 \pm 0.05)\,\eta$ or $(0.75 \pm 0.05)\,\eta$ for the PhO$^-$ DBS or 4-CP$^-$ QBS, respectively.

**The difference between DBS and QBS in the ponderomotive shift.** As the excess electron is still loosely bound in both DBS or QBS, its smaller slope (which is equivalent to the smaller effective polarizability) compared to the free-electron model seems to be quite reasonable. Namely, at the same intensity of the control laser pulse, the ponderomotive force on the quasi-free electron in DBS or QBS is smaller than that on the free-electron. This is in accord with the previously reported experiments on the atomic Rydberg orbitals[22,24,52]. Although the quantitative estimation of the effective polarizability seems to be a formidable task, it would be quite meaningful to inspect the difference between DBS and QBS in terms of the magnitudes of the ponderomotive forces on their excess electrons. As the slope of the 4-CP$^-$ QBS is much steeper than that of the PhO$^-$ DBS, the excess electron of the former could be considered to behave more like a free-electron compared to that of the latter. This seemingly makes sense as the electron binding energy of the 4-CP QBS ($\sim$20 cm$^{-1}$) is much smaller than that of the PhO$^-$ DBS ($\sim$97 cm$^{-1}$). That is, the loosely-bounded electron in the former is more polarizable compared to the relatively tighter-bounded electron in the latter. In order to testify this simple conjecture, we did carry out the similar experiment on 4-BP$^-$ as its DBS has the small electron binding energy of $\sim$24 cm$^{-1}$[53], which is comparable to that of 4-CP$^-$ QBS. Surprisingly, however, the $\Delta\tilde{\nu}:I$ slope of the 4-BP$^-$ DBS is found to be $(0.35 \pm 0.07)\,\eta$, Fig. 3, which is slightly larger than that of the PhO$^-$ DBS but it is still quite smaller than that of the 4-CP$^-$ QBS. The experimental fact that the effective polarizability of the 4-CP$^-$ QBS is much larger than that of the 4-BP$^-$ DBS, even though their electron binding energies are nearly same, indicates that the electron binding energy may not be the major distinguishing factor in dictating the effective polariz-abilities of NBS.

For the elaborate explanation for the difference of QBS and DBS, the pseudo-potential function of the monopole-quadrupole ($V_Q(r,\theta)$) or monopole-dipole ($V_\mu(r,\theta)$) interaction, respectively, has been invoked in the polar ($r,\theta$) coordinates where $r$ or $\theta$ is the radial distance or the polar angle with respect to the center of the quadrupole (or dipole), respectively[54].

$$V_Q(r,\theta) = -\frac{Q(3\cos^2\theta - 1)}{4r^3}\left[1 - \exp\left\{-\left(\frac{2r}{\sqrt{|Q|}}\right)^5\right\}\right] \qquad (4)$$

$$V_\mu(r,\theta) = -\frac{\mu\cos\theta}{r^2}\left[1 - \exp\left\{-\left(\frac{2r}{\mu}\right)^3\right\}\right] \qquad (5)$$

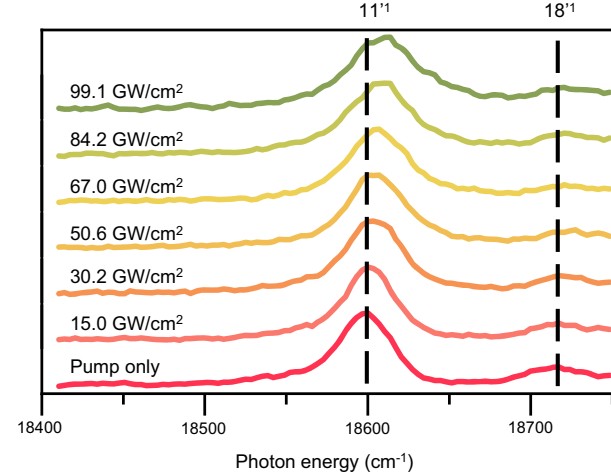

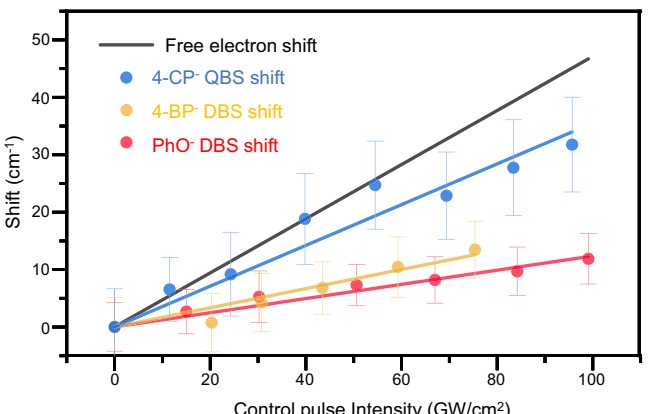

**Fig. 3 The (control laser) power-depenent ponderomotive shifts. a** Photodetachment spectra of the PhO$^-$ DBS at various intensities of the control laser pulse. Intensity of the control ps laser pulse for each spectrum is denoted. The most prominent $11'^1$ peak and adjacent $18'^1$ peak were labeled above the corresponding peaks (black dashed lines). **b** The blue-shift ($\Delta\tilde{\nu}$) of the PhO$^-$ DBS ($11'^1$, red circle), 4-CP$^-$ QBS ($12'^1$, blue circle), and 4-BP$^-$ DBS ($11'^1$, yellow circle) are plotted versus the intensity of the ps control laser pulse. The ponderomotive shift of the free-electron model from Eq. (3) (black solid line) is shown for the comparison. The linear fits to the experiment are given with experimental error bars ($\pm 1\sigma$), determined from the multiple measurements (>10) of the photodetachment spectra. The pump laser intensity was $\sim$10 GW/cm$^2$ for the 4-CP$^-$ QBS whereas it was $\sim$90 GW/cm$^2$ for the 4-BP$^-$ and PhO$^-$ DBSs. The same experiment with the much-reduced pump laser intensity ($\sim$10 GW/cm$^2$) has been carried out for the PhO$^-$ DBS, showing that the slope is barely influenced by the pump laser intensity (Supplementary Fig. 5). Source data are provided as a Source Data file.

here, $Q$ is the normalized quadrupole moment and $\mu$ is the dipole moment. Even though these potential functions may not exactly represent the templates for the quantum-mechanical non-valence orbitals of QBS or DBS, their characteristics in terms of the orbital shape could be approximately inferred. For the comparison of QBS and DBS having the similar electron binding energy, the above potential functions are visualized for the 4-CP$^-$ QBS or 4-BP$^-$ DBS by plugging the corresponding quadrupole or dipole moment into Eqs. (4) or (5), respectively (Fig. 4). Interestingly, according to these potentials, the non-valence orbital of the 4-CP$^-$ QBS is likely to be much more diffusive along both the radial and angular coordinates compared to that of the 4-BP$^-$ DBS. In the classical sense, the extent of the instant delocalization of the charged particle by the external oscillating electric field is

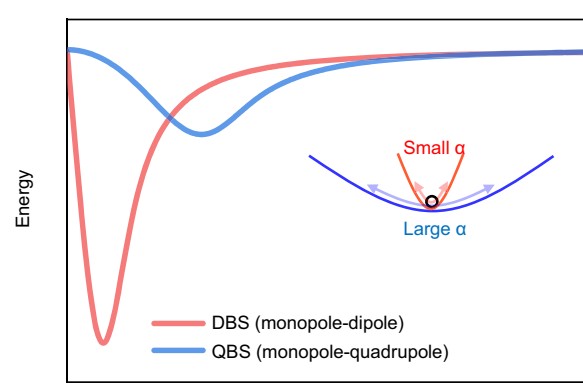

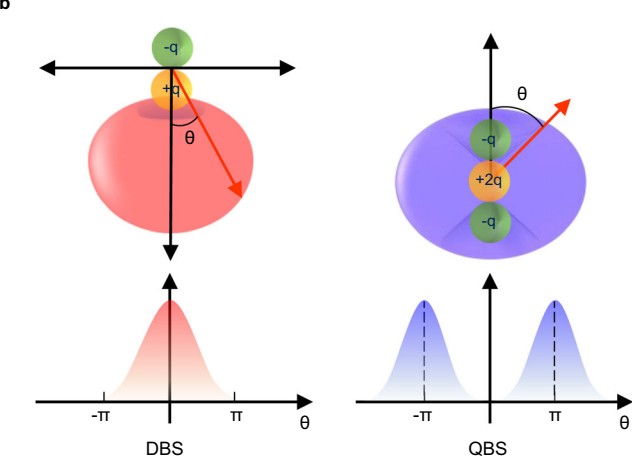

**Fig. 4 Long-range potentials and associated charge distributions of DBS and QBS. a** Long-range potential energy curves of the 4-BP⁻ DBS (monopole-dipole, red) and 4-CP⁻ QBS (monopole-quadrupole, blue) from the pseudo-potential functions given in Eqs. (4) and (5), respectively (see the text). Inset: the electron in the steep potential curve of the DBS exhibits the relatively smaller polarizability (small α) in the external oscillating field compared to that in the rather shallow potential curve of the QBS (large α). **b** The depicted (expected) angular distribution of the DBS (left) or QBS (right) orbital. The DBS electron (left) is confined in the narrow angular region at the positive side of dipolar molecule whereas the QBS electron is more widely distributed surrounding the molecular core.

also expected to be larger in the 4-CP⁻ QBS compared to that in the 4-BP⁻ DBS, as the force gradients with respect to the displacement of $r$, for example, is much gentler (steeper) in the former (latter) compared to the latter (former). In addition, considering the asymmetric electron charge distribution of DBS, we may envision the electron in DBS feels more the effect of the anionic core than that in QBS, and accordingly the free-electron-like character can be reduced in the case of DBS.

In summary, we have observed the ponderomotive effect on the quasi-free electron in the non-valence bound states of the polyatomic molecules. The entire DBS or QBS quantum states show the clear-cut blue-shifts and asymmetric broadening of which magnitudes are linearly proportional to the intensity of the non-resonant control laser pulse. It has been found that the non-valence electron behaves more like a free electron in the QBS compared to that in the DBS, indicating that the non-valence orbital of the QBS is more diffusive and thus more polarizable upon the external oscillating electromagnetic field. Our finding here strongly promises that the chemicophysical properties of the

polyatomic systems could be manipulated by utilizing the ponderomotive force on the quasi-free electron residing in the non-valence bound states, paving a new way for the laser control of polyatomic molecules.

## Methods

**Electrospray ionization-photoelectron imaging set-ups**. For sampling, 1 mM of phenol (>99.5%, TCI chemicals Inc.), 4-cyanophenol (>98.0%, TCI chemicals Inc.), or 4-bromophenol (>98.0%, TCI chemicals Inc.) was dissolved in the 9:1 methanol/water mixture without further purification. 15–20 drops of the 3 M ammonia/methanol solution were added in order to make the solution of pH ~ 9. The solution was sprayed into vacuum by a home-made electrospray ionization (ESI) assembly with the −3000 V ionization voltage. Anions were de-solvated and focused by a dual-stage ion funnel (IF141, MassTech Inc.) and transferred to a quadrupole ion trap (Jordan TOF Products Inc.) through the hexapole and octopole ion guides powered by the RF generators (Ardara Technologies L.P.). Before anions enter the ion trap, a quadrupole mass-filter was used in order to cut-off the low-mass solvent anions. Target anions were trapped and cooled for ~50 ms in the cryogenically-cooled ion trap coupled with 8 K He cryostat (Coolpower 10 MD, Leybold) by colliding with the 4:1 He:H₂ mixture of the buffer gas. The internally cooled anions were extracted from the ion trap and accelerated to the photoelectron velocity-map imaging (VMI) apparatus through the potential re-referencing tube. Anions are crossed by the picosecond laser pulses in the perpendicular geometry to emit photoelectrons. Those photoelectrons were accelerated to the position sensitive detector equipped with the chevron-type microchannel plates (MCP) backed by the P46 phosphor screen. The signals from the phosphor screen were recorded by the photomultiplier tube and transferred to the oscilloscope.

**Optical set-ups**. Picosecond laser pulses were generated from a 1 kHz repetition Ti:sapphire regenerative amplifier system (Legend Elite-P, Coherent) seeded by a femtosecond oscillator (Vitara-T, Coherent). Fundamental output (791 nm) from the regenerative amplifier was used to generate tunable-visible or UV light by an optical parametric amplifier (TOPAS-800, Light Conversion), which was used for the excitation (pump) pulse. Remaining fundamental output was used as the strong non-resonant pulse (control pulse). The delay between pump and control pulses was controlled by a DC-motor driven optical delay stage (DDS220, Thorlabs) combined with a retro-reflector (UBBR2.5-1UV, Newport). Laser intensities were measured using a set of laser power meter (FieldMaxII-TO, Coherent) and power sensor (PM10, Coherent). Beam waists (1/e²) of the pump or control pulses at the ion-laser interaction region were 385 μm or 890 μm, respectively. The pump pulse was more tightly focused than the control pulse in order to make the maximum number of ions influenced. Spatial and temporal overlaps between pump and control laser pulses were verified by monitoring the sum frequency generation (SFG, $\omega_{SFG} = \omega_{pump} + \omega_{control}$) signal from a beta-barium borate (BBO) crystal located outside the vacuum chamber. Both pump and control pulses were guided into the BBO crystal by a flip mirror. The BBO crystal was located so that both pulses are focused into the identical ion-laser interaction region. By adjusting the position of the focusing lenses (spatial overlap) and delay stage (temporal overlap), the maximum SFG intensity was achieved. The SFG signal was collected by a set of triangular prism and photodiode detector. Pulse duration of the both pump and control pulses was estimated to be ~1.7 ps. Cross-correlation of pump and control pulse was measured to be ~2.88 ps by SFG in a BBO crystal.

## Data availability

The datasets generated and/or analyzed during the current study are available within the paper as a source data. Source data are provided with this paper.

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

## Acknowledgements

This work was supported by the National Research Foundation of Korea under the Project Numbers of 2018R1A2B3004534 (S.K.K.), 2019K1A3A1A14064258 (S.K.K.), and 2019R1A6A1A10073887 (S.K.K.).

## Author contributions

D.H.K. and J.K. conducted whole experiments. D.H.K. and J.K. analyzed data. D.H.K. wrote the paper. H.R.N. and S.K.K. discussed the results. S.K.K. supervised the whole project and edited the manuscript.

## Competing interests

The authors declare no competing interests.
