## [Peer Review File · Nature Communications]

REVIEWER COMMENTS

Reviewer #1 (Remarks to the Author):

Report on:

Nature Communications: NCOMMS-21-33041

“Observation of the Ponderomotive Effect in Non-Valence Bound States of Polyatomic Molecular Anions”

Do Hyung Kang, Jinwoo Kim, Heung-Ryoul Noh, and Sang Kyu Kim

This paper describes the experimental observation of the Ponderomotive effect or dynamic Stark effect in the non-valence bound state of molecular anions. The DBS and QBS of the anions are prepared and controlled by picosecond laser pulses. By scanning the wavelength of the pump laser, the photodetachment spectra are obtained. With the control laser on or off, or by changing the intensity of the control laser pulse and its relative delay to the pump laser, the shift of the resonant peaks in the photodetachment spectra are observed, indicating a ponderomotive blue shift of DBS and QBS. Using the quasi-free electron model, the authors found the electron in the QBS is more susceptible to the external electromagnetic field compared to that in the DBS, suggesting that the QBS orbital is more diffusive and polarizable. This is a highly significant observation. It has long been known that the Ponderomotive effect or dynamic Stark effect induced by strong laser field affects the photodetachment threshold of atoms, molecules and ions, but this paper goes much further, directly measured the Ponderomotive shift of the non-valence bound anions. The conclusions are certainly correct. I think this paper merits publication in Nature Communications. However, there are a few points that the authors need to address properly, and present clearly in the manuscript before the article should be accepted.

1. Because the picosecond laser is used both for pump and control pulses, for better quantitative analysis, the field of the pump laser should be considered or is small enough to be ignored. Therefore, the intensity and size parameters of the two laser pulses should be presented.
2. Since ponderomotive blue shift is comparable to the linewidth of the picosecond laser, the energy shift determination in the photodetachment spectra is critical for the derivation of the definitive conclusions. The authors showed the details of the subtraction of direct detachment in photodetachment spectra in the supplementary materials (Fig. S3). They corrected the position of the stepwise direct detachment features as amount of the full-ponderomotive shift. In principle, the actual shift should be dependent on the spatial (focal sizes) and temporal (pulse durations) overlap of the two pulses. The authors should include more experimental details and explain the analysis process more clearly.

3. It is believed that the ponderomotive shift for the photodetachment threshold is under the influence of the ratio of the pump and control pulse durations (or intensity ratio of the overlapping part). Is this true for the non-valence bound electron as well in this study? Because only control laser pulse is used as shift-inducing field, and the observed ponderomotive blue shift is comparable to the linewidth of the picosecond pump laser, if the intensity of the pump laser barely influences the shift of the DBS/QBS, a pump laser with narrower linewidth (eg. a tunable nanosecond laser) should be used to increase the energy resolution, and clearly resolve the resonant peaks and detachment threshold for every vibrational channel. In that case, the conclusions would be more definitive, and more insights will be obtained.

4. The authors state that, in Fig. 2, the photodetachment spectra were obtained by monitoring only the low kinetic-energy electrons to avoid the contribution of the control laser pulse, which is very important to the reliable result derivation. Generally, in the direct detachment process, the kinetic energy of the photoelectrons will cover the energy from zero up to the limit of the total available energy (photon energy minus the binding energy). Due to the adiabatic (vertical) detachment nature of the non-valence bound electron, the vibrational state of the neutral core should keep unchanged when the electron is detached by the control laser pulse. Therefore, the photoelectron images and spectra at the studied resonant positions, with the control laser on and off respectively, need to be included in the supplementary materials, to illustrate that the signals from the control pulse do not show up at the low kinetic energy range.

Reviewer #2 (Remarks to the Author):

This manuscript describes a measurement that demonstrates the ponderomotive effect in non-valence bound states (dipole bound and quadrupole bound) of molecular anions for what is likely the first time. The effect is not huge, but the linear behavior, and the distinct behavior between dipole bound and the more diffuse quadrupole bound states is convincing. This work extends other important observations on the photophysics of molecular anions from the Kim group in recent times.

In general the manuscript is well-written and should be published with only minor revisions. Some specific comments:

page 2 line 7: please refine what is being expressed around '...acceleration of an electron/atom.'

Later in that paragraph, the sentence beginning 'And yet, ...' could be made more readable.

page 2 second-to-last line: 'Sine' -> 'Since'

In numerous places in the manuscript 'quadruple' is incorrectly used instead of 'quadrupole'.

Fig. 2(b) caption '...as the temporal overlap...'

in the methods section - perhaps the authors could clarify 'A few drops of the ammonia solution...'

Reviewer #3 (Remarks to the Author):

Basically, I liked this paper very much. The authors advanced a clever and rather novel idea about using the coupling between oscillating electromagnetic fields and diffuse electron states in anions to spatially manipulate them via the ponderomotive force. This contribution should be of great interest to your readers. My only problem with this paper was its rather incomplete referencing of work on dipole bound and quadrupole bound anions, especially of ground state ones, which are the ones of greatest practical importance. The following three papers should have been mentioned.

G. Liu, S. M. Ciborowski, J. D. Graham, A. M. Buytendyk, K. H. Bowen, J. Chem. Phys. 151, 101101 (2019).

G. Liu, S. M. Ciborowski, C. R. Pitts, J. D. Graham, A. M. Buytendyk, T. Lectka, and K. H. Bowen, Phys. Chem. Chem. Phys. 21, 18310-18315 (2019).

G. Liu, S. Ciborowski, J. Graham, A. Buytendyk, and K. Bowen J. Chem. Phys. 153, 044307 (2020).

Other than that, I strongly support the acceptance of this paper into Nature: Communication.

Reviewer #1 (Remarks to the Author):

This paper describes the experimental observation of the Ponderomotive effect or dynamic Stark effect in the non-valence bound state of molecular anions. The DBS and QBS of the anions are prepared and controlled by picosecond laser pulses. By scanning the wavelength of the pump laser, the photodetachment spectra are obtained. With the control laser on or off, or by changing the intensity of the control laser pulse and its relative delay to the pump laser, the shift of the resonant peaks in the photodetachment spectra are observed, indicating a ponderomotive blue shift of DBS and QBS. Using the quasi-free electron model, the authors found the electron in the QBS is more susceptible to the external electromagnetic field compared to that in the DBS, suggesting that the QBS orbital is more diffusive and polarizable. This is a highly significant observation. It has long been known that the Ponderomotive effect or dynamic Stark effect induced by strong laser field affects the photodetachment threshold of atoms, molecules and ions, but this paper goes much further, directly measured the Ponderomotive shift of the non-valence bound anions. The conclusions are certainly correct. I think this paper merits publication in Nature Communications. However, there are a few points that the authors need to address properly, and present clearly in the manuscript before the article should be accepted.

1. Because the picosecond laser is used both for pump and control pulses, for better quantitative analysis, the field of the pump laser should be considered or is small enough to be ignored. Therefore, the intensity and size parameters of the two laser pulses should be presented.

As the reviewer excellently has pointed out, the pump laser alone could induce the ponderomotive shift (as had been demonstrated for Rydberg states of atoms previously) (also shown in Figure S5, supplementary Information for the 11¹ mode of the PhO⁻ DBS). According to the reviewer's suggestion, we did present the detailed information of both pump and probe laser pulses in the revision (see the Figure 3 caption). Additionally, we have carried out the new experiment in order to see if the slope (the Stark shift *versus* the control laser-pulse intensity) is influenced by the intensity of the pump laser pulse. The pump laser intensity used for obtaining the slope of the DBS PhO⁻ presented in Figure 3 was fixed at ~ 90 GW/cm². In the new experiment in the revision process, the much-reduced pump-laser intensity of ~ 10 GW/cm² has been employed for the same purpose. We have found that the slopes of the Stark-shift *versus* the control laser pulse intensity obtained at the low (10 GW/cm²) and high (90 GW/cm²) pump laser intensities are nearly identical (see Figure S6, Supplementary Information), indicating that the ponderomotive shift by the control laser-pulse intensity is not influenced by the pump laser intensity in the wide dynamic range used in this work. **We included this observation in the revised caption of Figure 3.**

2. Since ponderomotive blue shift is comparable to the linewidth of the picosecond laser, the energy shift determination in the photodetachment spectra is critical for the derivation of the definitive conclusions. The authors showed the details of the subtraction of direct detachment in photodetachment spectra in the supplementary materials (Fig. S3). They corrected the position of the stepwise direct detachment features as amount of the full-ponderomotive shift. In principle, the actual shift should be dependent on the spatial (focal sizes) and temporal (pulse durations) overlap of the two pulses. The authors should include more experimental details and explain the analysis process more clearly.

Subtraction of the direct-detachment portion from the whole photodetachment spectra is not straightforward due to its slow-rising features despite that the blue-shift of the direct-detachment is quite obvious as manifested in Figure 1. The blue-shift of the prompt (<fs) direct-detachment process is supposed to follow the free-electron model, and thus for the unbiased analysis, all photodetachment spectra at many different intensities of the control laser-pulse have been obtained at the identical condition in terms of the spatiotemporal overlap between pump and control for all three anions. As the spatiotemporal overlap condition was kept same for both direct (continuum) and indirect (autodetachment) detachment process, our analysis (Supplementary Information Figure S3) seems to be valid in terms of the relative behaviors of the ponderomotive shifts of three different NBS with increasing the control laser intensity (Figure 3). However, the ponderomotive shift, as the reviewer correctly pointed out, is highly dependent on the spatiotemporal overlap of pump and control laser pulses, and thus the optical parameters are quite essential to reproduce the experiment. In the revised method section, therefore, we have presented the optical properties of pump and probe laser pulses in much more details (see the revised main 'method' section). Notably, as the vibrational Feshbach resonance bands quite stand out (especially for 11¹ band of PhO⁻ (or 4-BrPhO) or the 12¹ band of 4-CP⁻) compared to the direct-detachment background electron signal, the observed ponderomotive shift in the rather wide 0 – 30 cm⁻¹ range is not much influenced by the subtraction of the direct-detachment, giving the maximum error limit of ± 2 cm⁻¹ (revised Supplementary Information Figure S3).

3. It is believed that the ponderomotive shift for the photodetachment threshold is under the influence of the ratio of the pump and control pulse durations (or intensity ratio of the overlapping part). Is this true for the non-valence bound electron as well in this study? Because only control laser pulse is used as shift-inducing field, and the observed ponderomotive blue shift is comparable to the linewidth of the picosecond pump laser, if the intensity of the pump laser barely influences the shift of the DBS/QBS, a pump laser with narrower linewidth (e.g. a tunable nanosecond laser) should be used to increase the energy resolution, and clearly resolve the resonant peaks and detachment threshold for every vibrational channel. In that case, the conclusions would be more definitive, and more insights will be obtained.

The first part of the comment is true indeed, as demonstrated in the experiment done with changing the pump-probe delay time as illustrated in Figure 2. In this regard, the use of the *narrow band-width* pump laser, as the reviewer suggested, could be quite ideal in getting the better energy-resolution of the ponderomotive shift. One can thus conceive the combined use of nanosecond laser pulse (as a pump) and pico- (or femto-) second laser pulse (as a control) for the *better* energy-resolution and the *higher* peak-power, respectively. And yet, when the nanosecond (*pump*) and picosecond (*control*) laser pulses are employed, for instance, the fraction of the temporal overlapped (between the pump and control pulses) ensemble would be extremely small ($\sim 1/10^3$), and the meaningful experimental observation of the ponderomotive shift is expected to be less likely. However, we mentioned this in the *revised* text as one of the future possible works (on the bottom of page 7).

4. The authors state that, in Fig. 2, the photodetachment spectra were obtained by monitoring only the low kinetic-energy electrons to avoid the contribution of the control laser pulse, which is very important to the reliable result derivation. Generally, in the direct detachment process, the kinetic energy of the photoelectrons will cover the energy from zero up to the limit of the total available energy (photon energy minus the binding energy). Due to the adiabatic (vertical) detachment nature of the non-valence bound electron, the vibrational state of the neutral core should keep unchanged when the electron is detached by the control laser pulse. Therefore, the photoelectron images and spectra at the studied resonant positions, with the control laser on and off respectively, need to be included in the supplementary materials, to illustrate that the signals from the control pulse do not show up at the low kinetic energy range.

The reviewer's comment is completely right. The photoelectron spectra taken at the resonant position (the 11^1 band of PhO⁻ DBS) *with* and *without* the control laser pulse are *identical* (namely, the ponderomotive shift does not affect the photoelectron peaks, reflecting the resonant nature of the NBS) and do not show any high-energy peaks due to the control laser pulse (in the revised Supplementary Information Figure S7). In the photoelectron spectra taken at non-resonant position where the direct-detachment prevails, on the other hand, the significant red-shifts of the photoelectron bands are clearly observed in the presence of the control laser-pulse compared to those taken without the control laser pulse, verifying the ponderomotive blue-shift of the direct-detachment threshold by the control laser pulse (see the revised Supplementary Information Figure S8).

Reviewer #2 (Remarks to the Author):

This manuscript describes a measurement that demonstrates the ponderomotive effect in non-valence bound states (dipole bound and quadrupole bound) of molecular anions for what is likely the first time. The effect is not huge, but the linear behavior, and the distinct behavior between dipole bound and the more diffuse quadrupole bound states is convincing. This work extends other important observations on the photophysics of molecular anions from the Kim group in recent times.

In general the manuscript is well-written and should be published with only minor revisions. Some specific comments:

page 2 line 7: please refine what is being expressed around '...acceleration of an electron/atom.'

Later in that paragraph, the sentence beginning 'And yet, ...' could be made more readable.

page 2 second-to-last line: 'Sine' -> 'Since'

In numerous places in the manuscript 'quadruple' is incorrectly used instead of 'quadrupole'.

Fig. 2(b) caption '...as the temporal overlap...'

in the methods section - perhaps the authors could clarify 'A few drops of the ammonia solution...'

According to the reviewer's comments, we have corrected several typos and improper expressions in the revision. Some sentences and references, which had been expressed unclearly, have been refined now. We appreciate the reviewer's comments and suggestions.

Reviewer #3 (Remarks to the Author):

Basically, I liked this paper very much. The authors advanced a clever and rather novel idea about using the coupling between oscillating electromagnetic fields and diffuse electron states in anions to spatially manipulate them via the ponderomotive force. This contribution should be of great interest to your readers. My only problem with this paper was its rather incomplete referencing of work on dipole bound and quadrupole bound anions, especially of ground state ones, which are the ones of greatest practical importance. The following three papers should have been mentioned.

G. Liu, S. M. Ciborowski, J. D. Graham, A. M. Buytendyk, K. H. Bowen, J. Chem. Phys. 151, 101101 (2019).

G. Liu, S. M. Ciborowski, C. R. Pitts, J. D. Graham, A. M. Buytendyk, T. Lectka, and K. H. Bowen, Phys. Chem. Chem. Phys. 21, 18310-18315 (2019).

G. Liu, S. Ciborowski, J. Graham, A. Buytendyk, and K. Bowen J. Chem. Phys. 153, 044307 (2020).

Other than that, I strongly support the acceptance of this paper into Nature: Communication.

Suggested references have been included in the manuscript properly. As the reviewer suggested, the ground state DBS and QBS are stable and they may possess the great potential on the quantum manipulation that could be utilized by the ponderomotive effect in near future.

REVIEWERS' COMMENTS

Reviewer #1 (Remarks to the Author):

The authors have addressed my questions well. I am satisfied with the revised paper.